# Contrastive Explanations of Centralized Multi-agent Optimization Solutions

**Primary Keywords:** *Human-aware Planning and Scheduling*

## Abstract

In many real-world scenarios, agents are involved in optimization problems. Since most of these scenarios are overconstrained, optimal solutions do not always satisfy all agents. Some agents might be unhappy and ask questions of the form "Why does solution $S$ not satisfy property $P$?". We propose CMAoE, a domain-independent approach to obtain *contrastive explanations* by: (i) generating a new solution $S'$ where property $P$ is enforced, while also minimizing the differences between $S$ and $S'$; and (ii) highlighting the differences between the two solutions, with respect to the features of the objective function of the multi-agent system. Such explanations aim to help agents understanding why the initial solution is better in the context of the multi-agent system than what they expected. We have carried out a computational evaluation that shows that CMAOE can generate contrastive explanations for large multi-agent optimization problems. We have also performed an extensive user study in four different domains that shows that: (i) after being presented with these explanations, humans' satisfaction with the original solution increases; and (ii) the constrastive explanations generated by CMAoE are preferred or equally preferred by humans over the ones generated by state of the art approaches.

## Introduction

In many real-world scenarios, centralized AI systems generate solutions for optimization problems involving multiple agents with conflicting preferences. Due to these conflicts and the over-constrained nature of the problems, satisfying all agents' preferences is often impossible, and AI decisions might lead to some agents being unhappy (Kraus et al. 2020). In such situations, it is natural that some agents question the decisions made by the AI system, since there is a mismatch between the proposed solution and the user's mental model (Chakraborti et al. 2017).

Generating explanations for such questions may improve the AI system's transparency, facilitate human-computer collaboration, and increase human satisfaction (Bradley and Sparks 2009). Studies in different areas ranging from Explainable AI (Krarup et al. 2021) to social sciences (Lim, Dey, and Avrahami 2009; Miller 2019) or marketing (Tomaino et al. 2022) show that users typically formulate "why?" questions when they are not happy with a given solution. These questions usually take the form of

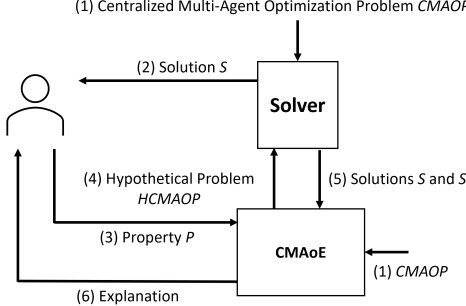

Figure 1: Schematic representation of CMAOE.

"Why does solution $S$ not satisfy property $P$?". There exist two main approaches to answering these questions in the literature: *counterfactual* and *contrastive* explanations.

Counterfactual explanations (Wachter, Mittelstadt, and Russell 2017; Korikov, Shleyfman, and Beck 2021) try to provide explanations by constructing a hypothetical situation where the agent would have received its desired solution if its inputs were different. These explanations take the form of "solution $S$ would have satisfied property $P$ if your input to the system had been $Y'$ rather than $Y''$". This type of explanations are not always valid, as the suggested recourses might not be actionable in practice. Another alternative to answering "why?" questions focuses on *contrastive explanations* (Lipton 1990; Miller 2021). In this case, the focus is purely on understanding why the solution returned by the AI system was a better choice than the one the human agent who requested the explanation had in mind. Unlike counterfactuals, contrastive explanations do not prescribe actions, so they can be safely generated across a wider range of domains.

Inspired by (Krarup et al. 2021), in this paper we present CMAoE, a domain-independent approach to generate local contrastive explanations for multi-agent optimization problems where decision making is centralized. Figure 1 depicts CMAoE. First, the solver generates a solution $S$ for the original Centralized Multi-Agent Optimization Problem (CMAOP) and presents it to the user. Then, the user formulates a question asking why solution $S$ does not satisfy property $P$. This is passed to CMAoE, which generates a

new hypothetical CMAOP (HCMAOP) that forces property $P$ to be satisfied. The hypothetical problem suggested by the user might lead to a new solution that is very different from the original one. This is often not desirable, since (i) the new solution should be plausible in the real world, i.e., it should be similar to the one generated by the solver; and (ii) a large number of changes between the original and new solutions would entail longer explanations that could frustrate users. Therefore, HCMAOP not only encapsulates property $P$, but also modifies the original problem to minimize the number of changes between the original solution $S$ and the new solution $S'$. The Solver sends these two solutions to CMAoE, which finally generates an explanation by computing their differences. This process is iterative, as the user can ask further questions until the explanation is satisfactory, also gaining a better understanding of the decision-making process followed by the Solver at each iteration.

The main contributions of this paper are: (i) the definition of hypothetical CMAOP; (ii) the automated generation and solving of HCMAOP that yields contrastive explanations; (iii) a computer-based evaluation of several CMAOP tasks; and (iv) an extensive user-study in four domains with more than 240 users that shows the value and user acceptance of those explanations, as well as their preference of contrastive over counterfactual explanations.

The rest of the paper is organized as follows. We first formalize CMAOPs and introduce a novel approach to generate contrastive explanations of solutions to these problems. We evaluate CMAoE on four well-known CMAOPs, showing how it generates explanations in a similar time as the one needed to solve the original problem. Later, we present the results of an extensive user study in four well-known CMAOPs with more than 240 users. After being presented with these explanations, humans' satisfaction with the original solution increases and their desire to complain decreases. Furthermore, the explanations generated by CMAoE are preferred or equally preferred by humans over the ones generated by state of the art approaches. Finally, we draw our main conclusions and outline future work.

## Centralized Multi-Agent Optimization Problems

Centralized Multi-Agent Optimization Problems (CMAOPs) are solved by finding an optimal solution that minimizes (or maximizes) a given objective function from a set of alternatives, taking into account a set of constraints referring to a set of agents. We focus on CMAOPs where the decision-making is centralized, i.e., a central entity solves the optimization problem by considering the agents' preferences and constraints. Many real-world scenarios lie under this framework, such as nurse shift scheduling, task or resource allocation, or logistics planning. CMAOPs are formally defined as follows:

**Definition 1** *A **Centralized Multi-Agent Optimization Problem (CMAOP)** is a tuple CMAOP $= \langle A, X, C, f, m \rangle$ where $A$ is a set of agents, $X$ is a domain of feasible points subject to set of constraints $C$, $f$ is the objective function, and $m$ is the goal function which is either min or max.*

The Mixed Integer Programming (MIP) formulation of CMAOPs is based on an algebraic specification of a set of feasible alternatives, as well as the objective criterion for comparing alternatives. This is achieved by: (i) introducing discrete and/or continuous decision variables; (ii) expressing the criterion as a function of variables; and (iii) representing the set of feasible alternatives as the solutions to a conjunction of constraints described as equations and inequalities over the variables. Therefore, MIP provides a general framework for modeling a large variety of CMAOPs such as scheduling, planning, task assignment, network design, etc. A general MIP formulation of a CMAOP is defined below:

$$
\begin{aligned}
\min \quad & f(x) \\
\text{s.t.} \quad & g_i(x) \geq 0 \quad i = 1, ..., q \\
& h_j(x) = 0 \quad j = 1, ..., p
\end{aligned}
$$

where $x \in X \subseteq \mathbb{R}^n$ represents an n-variable decision vector subject to a feasible set $X$. Let $a(x) \subseteq x$ refer to the subset of decision variables involving agent $a \in A$. One decision variable $x$ could involve multiple agents. The objective function $f(x)$ is composed of two sub-terms: $f_A(x)$, a function over the subset of decision variables involving agents' inputs; and $f_G(x)$, a function over the subset of decision variables not involving any agents' input.

The feasible set $X$ is given by the set of constraints in $C$. These constraints are represented as inequalities ($g_i$) or equalities ($h_j$) of functions over the decision vector $x$.

Finally, a maximization problem can be modelled as a minimization problem by multiplying the objective function by (-1). We denote the quality of a solution to a CMAOP as $q\big(f(x)\big)$. An optimal solution is represented as $x^*$. In the rest of the paper, we will always refer to optimal solutions when we talk about solutions of CMAOPs.

**Running Example: the Knapsack Problem** Let us introduce a Knapsack Problem (KP) as the running example throughout the paper. In KP, each agent from a set of agents owns a few items. Each item occupies a different space, and each agent assigns a different utility value to each item, i.e., how much they appreciate their items. The agents share a common depot with a limited capacity where items can be included. The problem is determining the items to be included in the depot so that the total utility is maximized, while satisfying the depot's capacity and considering some fairness issues. A formal MIP formulation of a KP is shown below. The objective function to optimize is:

$$
\max \sum_{a \in A, i \in I} x_{a,i} \times \text{UTILITY}(a,i) + \text{minItems} \tag{1}
$$

subject to the following constraints:

$$
\sum_{a \in A, i \in I} x_{a,i} \times \text{SPACE}(i) \leq \text{depotCapacity} \tag{2}
$$

$$
\sum_{i \in I} x_{a,i} \geq \text{minItems} \tag{3}
$$

$$
x_{a,i} \in \{0, 1\} \tag{4}
$$

$$
\text{minItems} \in \mathbb{Z} \tag{5}
$$

There is one binary decision variable $x_{a,i}$ for each agent $a \in A$ and item $i \in I$. These variables will take a value

of 1 if item $i$ from agent $a$ is included in the depot. Integer decision variable minItems keeps track of the number of items belonging to the agent with the least number of items included in the depot (Constraint 3). The objective function maximizes the utility of the included items, and the number of items belonging to the agent with the least number of items included in the depot. Constraint 2 ensures that the maximum capacity of the depot is not exceeded.

In this case, the first term of the objective function corresponds to $f_A(x)$, i.e., variables explicitly associated with agents' inputs, while the second term corresponds to $f_G(x)$, i.e., other variables not involving agents' inputs.

# CMAoE: Generating Contrastive Explanations

Given that most CMAOPs are over-constrained, optimal solutions do not always satisfy all agents, who might be unhappy and ask questions about the solution. In this paper, we focus on generating local contrastive explanations for questions that take the form of "Why does solution $S$ not satisfy property $P$?". The next sections describe how CMAoE works: (i) it builds a hypothetical CMAOP; and (ii) it generates explanations from the comparison of the solutions returned by the original and the hypothetical CMAOP.

## Building a Hypothetical CMAOP

To generate these explanations, we build a hypothetical optimization problem where property $P$ is forced to be satisfied. This hypothetical problem might generate a solution that is very different from the original one. To prevent this, we further modify the original problem to minimize the number of changes between the original and new solution, $S$ and $S'$.

We first define how to compute the differences between two solutions. In many scenarios, the set of decision variables $x$ used to model CMAOPs includes some auxiliary variables that are used to help the modeling process or improve the efficiency when solving the problem. We refer to the subset of decision variables in a CMAOP that actually represent a solution as $S \subseteq x$. These are the variables considered when computing the differences between two solutions. Going back to our KP running example, while $x_{a,i}$ and minItems are the decision variables of the problem $(x)$, only $x_{a,i}$ variables would be in $S$, since they are the only ones representing a solution.

We formally define a Hypothetical Multi-Agent Optimization Problem (HCMAOP) as an extension of the original CMAOP as follows:

**Definition 2** *A **Hypothetical Multi-Agent Optimization Problem** (HCMAOP) is a tuple HCMAOP = $\langle A, X', C', f, m, S, P \rangle$, where $A$ is a set of agents, $X'$ is an updated domain of feasible points subject to the new set of constraints $C'$, $f$ and $m$ remain as in the original OP, $S$ is a solution to the original CMAOP, and $P$ is the hypothetical property to be enforced.*

We extend the original set of constraints $C$ with a new set of constraints that enforce that the hypothetical property $P$ is satisfied. From now on, we assume HCMAOP is solvable,

i.e., enforcing the hypothetical property $P$ might affect quality but not solvability. We also extend the original decision vector $x$ with a new set of decision variables $z(S)$ that will compute the differences between the solution to the original CMAOP $(S)$ and a solution to the new HCMAOP $(S')$. The value of these new variables is ensured by a set of constraints $Z$, i.e., $C' = C \cup P \cup Z$. Finally, we also modify the original objective function to reason at the same time about the quality of the solution, $f(x)$, and the number of changes between the original and the hypothetical solution, $z(S)$.

A general MIP formulation of a HCMAOP is defined as:
$$\begin{aligned} \min \quad & \alpha f(x') + \beta z(S) \\ \text{s.t.} \quad & C' \end{aligned}$$

where $x' \in X'$ represents the new decision vector subject to the feasible set $X'$. We introduce two parameters (weights) in HCMAOP's objective function, $\alpha$ and $\beta$. By modifying these parameters, we will generate solutions that prioritize either maximizing the quality of the hypothetical CMAOP solution or minimizing the number of changes between the original $(S \subseteq x)$ and the hypothetical $(S' \subseteq x')$ CMAOP solutions. In scenarios where we maximize quality, the solution to the HCMAOP problem might be arbitrarily different from the original solution, thus yielding very long explanations. However, minimizing the number of changes between the original $(S \subseteq x)$ and the hypothetical $(S' \subseteq x')$ CMAOP solutions leads to shorter and more concise explanations.

**KP running example**. Now, agent "Alice" asks why item "bed" has not been included in the optimal CMAOP's solution $x^*$. We build the associated HCMAOP as follows. First, we enforce the agent's property by adding the following constraint, which ensures that the bed is included in the depot. Then we generate a new set of $z(S)$ variables by duplicating the decision variables used to represent the original solution. In this case, we add one $z_{a,i}$ variable for each original $x_{a,i}$ variable. We also add the following constraints to ensure that the $z$ variables capture the changes of the new solution with respect to the original one:
$$x'_{a,i} - x_{a,i} \leq z_{a,i}, \qquad x_{a,i} - x'_{a,i} \leq z_{a,i}, \qquad z_{a,i} \geq 0$$

Finally, we update the objective function:
$$\max \alpha \Big( \sum_{a \in A, i \in I} x_{a,i} \times \text{UTILITY}(a,i) + \text{minItems} \Big) - \beta \sum_{a \in A, i \in I} z_{a,i}$$

In the following, we focus on two main variations of CMAoE: Q-CMAoE, which prioritizes quality by making $\alpha \gg \beta$; and C-CMAoE, which prioritizes the number of changes by making $\beta \gg \alpha$. Both variations still reason about both terms.

The complexity of the new HCMAOP compared to that of the original CMAOP relates to how many new constraints and variables need to be added. The number of constraints and variables to add depends on (i) the number of decision variables used to represent a solution in the original CMAOP; and (ii) the hypothetical property $P$ to be enforced. HCMAOP's objective function is more complex since it also reasons about $z(S)$ variables. However, as we will see later, the hypothetical property $P$ tends to constrain the solution space so that HCMAOPs can be efficiently solved in practice.

## Generating Explanations from CMAOP and HCMAOP Solutions

The next step of CMAoE is to generate explanations by computing the differences between the solution to the original CMAOP $S$, and the solution to the HCMAOP $S'$. We envision generating two types of explanations, depending on the level of abstraction we want.

**Abstract Explanation**. The explanation only refers to the difference in the quality of both solutions.

$$\text{Quality Diff} = q\big(f(x)\big) - q\big(f(x')\big) \qquad (6)$$

**Full Explanation**. The explanation refers to the specific changes between $S$ and $S'$, grouping them by agents. Algorithm 1 outlines how this computation is performed. The

---

**Algorithm 1: Full Explanation Generation**

**Require:** HCMAOP $= \langle A, X', C', f, m, S, P \rangle, S'$
**Ensure:** $\mathcal{E}$
1: $\mathcal{E}_\mathcal{I} \leftarrow \emptyset$ , $\mathcal{E}_\mathcal{D} \leftarrow \emptyset$
2: $\mathcal{I} = \textsc{computeIncreases}(S, S')$
3: $\mathcal{D} = \textsc{computeDecreases}(S, S')$
4: **for** $a \in A$ **do**
5: $\quad \mathcal{I}_a = a(\mathcal{I}), \mathcal{D}_a = a(\mathcal{D})$
6: $\quad \mathcal{E}_\mathcal{I} \leftarrow \mathcal{E}_\mathcal{I} \cup \{\langle i, q(f(i)) \rangle \mid i \in \mathcal{I}_a\}$
7: $\quad \mathcal{E}_\mathcal{D} \leftarrow \mathcal{E}_\mathcal{D} \cup \{\langle d, q(f(d)) \rangle \mid d \in \mathcal{D}_a\}$
8: **end for**
9: $\mathcal{E} \leftarrow \mathcal{E}_\mathcal{I} \cup \mathcal{E}_\mathcal{D}$
10: **return** $\mathcal{E}$

---

algorithm receives as input the HCMAOP and the solution to the new problem (HCMAOP already contains the solution to the original problem). First, the algorithm computes the subset of decision variables representing a solution whose value has either increased or decreased between the original and the hypothetical solution. This is done by the COMPUTEINCREASES and COMPUTEDECREASES functions, which return the set of increase ($\mathcal{I}$) or decrease ($\mathcal{D}$) changes, respectively. Then, the algorithm iterates over the set of agents, computing the subset of increases and decreases where agent $a$ was involved (line 5). After that, it computes the contribution of these changes to the objective function and updates the sets of increases and decreases (lines 6 and 7). Finally, the algorithm returns the explanation $\mathcal{E}$, which is the union of all the decision variables in the solution that either increased or decreased their value in the new solution $S'$ with respect to the original solution $S$.

**KP running example**. Let us assume that the only change between solutions $S$ and $S'$ is Bob removing his bed (with a utility of $4$) in favour of Alice's bed (with a utility of $2$) increasing The Abstract Explanation would be that doing so would mean a loss of 2 utility units. The Full Explanation would be: $\mathcal{E}_\mathcal{I} = \{\langle x_{\text{Alice,bed}}, 2 \rangle\} \cup \mathcal{E}_\mathcal{D} = \{\langle x_{\text{Bob,bed}}, 4 \rangle\}$.

## Computational Evaluation

Although our formalization is general to any CMAOP described as a MIP, we run experiments using Mixed Integer Linear Programs (MILP) for which optimal solutions are easier to compute. We evaluate our approach by providing explanations in simulated scenarios in four well-known CMAOPs that can be formulated as MILPs. Below, we provide a brief description of each domain. A formal definition of their associated MILPs can be found in Appendix A.

- **Knapsack Problem (KP)**. A variation of our KP running example where the objective function only optimizes the total utility of the items included in the depot.

- **Task Allocation Problem (TAP)**. A set of tasks needs to be allocated to a set of agents. Each agent has a maximum workload. Each agent assigns a different utility value to each task. The goal is to assign all the tasks, while maximizing the total utility of the assignment and respecting agents' workload.

- **Wedding Seating Problem (WSP)**. A set of agents need to be seated at a set of tables with different capacities. Each pair of agents has an associated affinity value, i.e., how much they would like to be seated at the same table. The AI system's problem is determining the allocation of agents to tables so that the total affinity is maximized while satisfying the tables' capacities.

- **Capacitated Vehicle Routing Problem (CVRP)**. A set of agents (vehicles) with heterogeneous capacities have to visit a set of points distributed on a map. The number of points a vehicle can visit is given by its capacity. All agents start and end in a depot. Each pair of points has an associated distance. The AI system needs to determine the route of vehicles so that the total traveled distance is minimized while satisfying the vehicles' capacities.

## Experimental Setting

We have generated problems in the four domains by fixing some inputs and general constraints of the problem: the depot's capacity in KP, the tables' capacity in WSP, the number of points and vehicles' capacity in CVRP, and the number of agents in TAP. For each configuration, we have generated 10 CMAOPs with random utilities/affinities/distances depending on the domain. We optimally solved each problem and automatically computed all of the unsatisfied variables, i.e., those decision variables representing the solution with a value of zero. Then, we randomly picked 10 of these unsatisfied variables in order to generate 10 hypothetical properties, i.e., agents' questions about the original solution. For example, in KP we get all of the items that were not included in the depot and generate 10 questions of the form *"why was item x not included in the depot?"*. This yielded 100 HCMAOPs to solve. We can solve each problem with either of the two variations of our approach: Q-CMAoE or C-CMAoE. For each problem, we report (i) the time needed to compute the solution for the CMAOP and the HCMAOP; (ii) the quality of the CMAOP and the HCMAOP; and (iii) the length of the explanation, i.e., the number of changes between the CMAOP and HCMAOP solutions.

MILP problems were modeled using the PuLP Python library (Mitchell, OSullivan, and Dunning 2011) and solved using the CBC solver (Forrest and Lougee-Heimer 2005). Experiments were run in Intel(R) Xeon(R) CPU E3-1585L v5 @ 3.00GHz machines with 64GB RAM and a 60s timeout (including the solving and model-building times).

## Scalability Evaluation

We first evaluate the scalability of our approach by comparing the time (in seconds) needed to compute the original and the hypothetical solutions as we increase the complexity of the problem. In KP, we increase the complexity of the problem by increasing the number of agents and setting the depot's capacity to be a function of this number. In TAP, we fix the number of agents and increase the number of tasks to be assigned. In WSP, we also increase the number of agents, fixing the number of tables but varying their size to be able to accommodate all agents. Finally, in CVRP, we increase the number of points, fixing the number of vehicles but varying their capacity to be able to visit all points. All of these problems are solved using Q-CMAoE. Execution times are similar for C-CMAoE.

The results of this experiment are shown in Figure 2. As expected, increasing the complexity of the problems leads to longer solving times and explanation generation times in all domains. However, while generating explanations is notably faster than solving the problem in WSP and CVRP, the opposite occurs in KP and TAP. This is because these problems only have one type of decision variable and only a few types of constraints, while the other domains involve more constraints and interrelated decision variables.

We conclude that the time needed to generate an explanation compared to the time needed to generate a solution will vary depending on the problem to be solved and its formulation. However, this time difference does not usually exceed an order of magnitude and remains constant as problems become more complex. Therefore, our approach will generate explanations for any MILP for which a solution can be generated within reasonable time and memory bounds.

## Solution Quality vs Explanation Length Trade-off

We analyze the trade-off between quality and explanation length by comparing our two approaches and measuring: (i) Explanation Length —the number of changes between the original and hypothetical solutions, $|\mathcal{E}|$; and (ii) Suboptimality Ratio —the quality of the original vs. the hypothetical solution, $q(f(S))/q(f(S'))$. The results of this experiment are shown in Figure 3 for the smaller problems in all domains, i.e., 10 agents in KP, 10 tasks in TAP, 8 agents in WSP and 8 points in CVRP. Conclusions drawn from bigger problems are the same, and their results are shown in Appendix B.

As expected, optimizing the quality of solution $S'$ first (Q-CMAoE) yields solutions closer to the optimal one $S$ (lower Suboptimality Ratios) at the expense of generating a new solution $S'$ with more changes (higher Explanation Length). On the other hand, minimizing the number of changes between $S$ and $S'$ first (C-CMAoE), generates shorter explanations at the expense of having slightly worse solutions in terms of quality. Despite this trend, we often get the same Suboptimality Ratio and Explanation Length regardless of the approach used. From 100 problems generated for each domain, we obtain the same values in 67 problems in KP, 30 in TAP, 39 in WSP, and 9 in CVRP. When focusing on problems for which we obtain similar rather than exact values, there are 77 problems in KP, 30 in TAP, 49 in WSP, and 33 in CVRP for which the Suboptimality Ratio and Explanation Length values differ by less than 30%. These results show that, in most cases, there are no big differences in the solutions produced by both approaches.

## User Study 1: CMAoE Validation

We designed and implemented a between-subjects user study to validate CMAoE. In the following subsections, we detail the setup, results and analysis.

### Setup

We generated **abstract** and **full** explanations using Q-CMAoE. We chose this variation given the similar results both approaches got in the previous section, while Q-CMAoE generates solutions with slightly higher quality. We compared these explanations against a **baseline** explanation. The objective of the baseline is to demonstrate that our explanation was the determining factor in the users' behavior, rather than simply providing any explanation. This is necessary based on the findings in (Kosch et al. 2023) that demonstrated the placebo effect in AI experiments. In particular, in this user study, we used "Sorry, this is what the algorithm generated" as a baseline explanation. We wanted to validate the following hypotheses:

**Hp1**: CMAoE's explanations improve humans' satisfaction with the decisions of the AI system.

**Hp2**: CMAoE's explanations reduce humans' desire to complain about the decisions of the AI system.

**Hp3**: Humans prefer more detailed explanations.

This experimental setting just focuses on assessing whether our contrastive explanations improve humans' satisfaction and decrease their desire to complain or not.

The evaluation of CMAoE against other state of the art approaches is discussed in detail in User Study 2.

By considering the four domains discussed in previous section and the three types of explanation, we generated twelve scenarios. We force the original solution to be very unfavourable for the participants so they have a reason to complain. In each scenario, participants were asked (i) their *user satisfaction* for the solution generated by AI, and (ii) their *desire to make a complaint* regarding that solution. The answers to both questions were measured on a 5-point Likert scale, where 1 represents the lowest and 5 is the highest. Then, in each scenario, regardless of the level of satisfaction or their desire to complain, the participants were presented with one of the three different types of explanations.

Afterward, the same set of questions was repeated in order to compare the participants' satisfaction and desire to complain before and after receiving the explanation. Each user was asked to rate their satisfaction with the explanation on a Likert scale. Finally, we asked a domain-related question to verify the users' comprehension of the domain; e.g. in KP, asking what the goal of the AI algorithm was.

We recruited 207 computer science students, 75 females and 132 males. The average age was 24.75 (std=3.54). Each participant was shown 2 or 3 scenarios randomly, ensuring that no domain or explanation type was repeated. We discarded 34 scenarios where users answered the verification

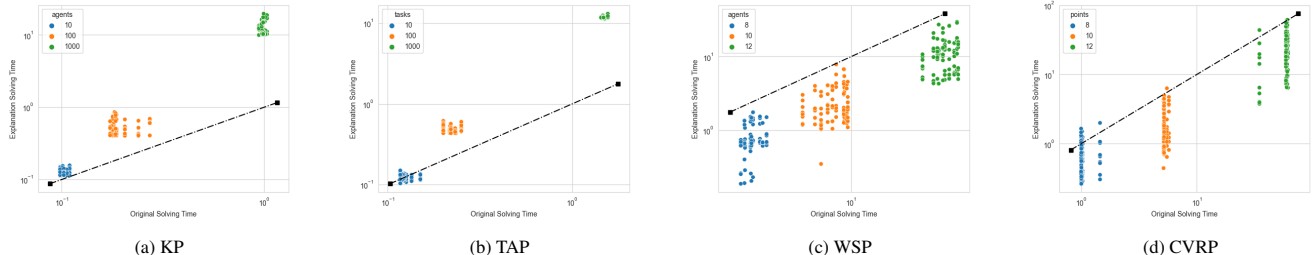

(a) KP      (b) TAP      (c) WSP      (d) CVRP

Figure 2: Time (seconds) in log scale needed to compute the original and hypothetical CMAOP as we increase the complexity of the problem. Points below the diagonal line represent problems for which producing the explanation takes less time than solving the original problem.

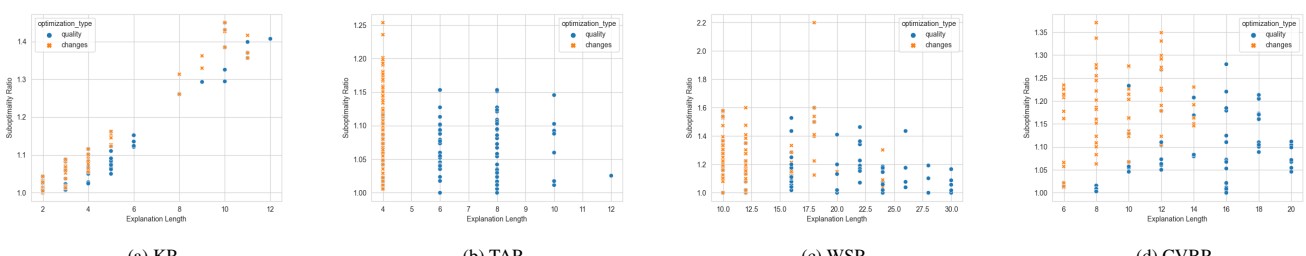

(a) KP      (b) TAP      (c) WSP      (d) CVRP

Figure 3: Trade-off between Explanation Length and Suboptimality Ratio when solving the HCMAOP with Q-CMAoE (blue points) or C-CMAoE (orange crosses).

question incorrectly, leading to 100, 85, 76 and 84 scenarios examined for the KP, WSR, CVRP and TAP domains, respectively. We used repeated measures ANOVA (General Linear Model) for each domain separately.

## Human Satisfaction & Desire to Complain

In the first part of the user study, we evaluate $Hp_1$ and $Hp_2$. Table 1 shows the number of users ($N$) who participated in each scenario (distinct pairs of domain and explanation type). Further, it presents the mean and standard deviation of user satisfaction with the solution and their desire to complain. Based on the table, in all domains, the initial satisfaction of the users with the solution was similar among all users with different explanations ($p > 0.05$). This is a good indication of our randomized distribution of the users between all sessions since, at the initial stage, no explanation was presented to the users.

From a post-hoc analysis with correction for multiple measurements in all domains, the level of satisfaction after receiving the explanation is still low. This is because the original solution was very unfavourable for the participants, and CMAoE just tries to explain the outcome rather than changing it. Despite the general low satisfaction level, we can observe that the increase in user's satisfaction level was significantly greater following the *abstract* and *full* explanations compared to the *baseline* explanation. For more details about the *main* and *interaction* effects, an ANOVA analysis is described in Appendix D.

The results presented in this table confirm $Hp1$, i.e. the explanations generated by CMAoE improve humans' satisfaction regarding decisions made by the AI system.

Finally, from a post-hoc analysis with correction for multiple measurements in all domains, the decrease in desire to complain was significantly greater following the *abstract* and *full* explanations compared to the *baseline* explanation. Similarly, an ANOVA analysis of the results is shown in Appendix D. The results presented in this table confirm $Hp2$, that the generated explanations by CMAoE decrease humans' desire to complain regarding decisions made by AI.

## Length of Explanation

In this part of the study, we validate our third hypothesis, $Hp_3$. The results in Table 2 present a between-subjects analysis comparing the satisfaction between the users in each of the explanation types, using Univariate Analysis of Variance. In all domains, we found an effect for the type of explanations on user satisfaction. The F-score and p-value were; KP($F(2) = 19.52$, $p \ll 0.01$), WSP ($F(2) = 14.62$, $p \ll 0.01$), CVRP ($F(2) = 10.64$, $p \ll 0.01$) and TAP ($F(2) = 6.9$, $p < 0.01$). In general, in all domains, users reported higher satisfaction with the *abstract* and *full* explanations rather than the *baseline* explanation. In CVRP and WSP, users were significantly more satisfied with the *full* explanation in comparison to the *abstract* explanation. However, their satisfaction with the abstract and full explanations was much closer in the KP and TAP domains. We hypothesize this is because KP and TAP are much simpler domains in comparison to CVRP and WSP. The results of Table 2 confirm $Hp3$ —users prefer a more detailed expla-

Table 1: Satisfaction and desire to complain ('mean (std)') about the original solution before and after participants are presented with CMAoE's explanations in the four domains. $N$ represents the number of participants per session.

| Status | Exp. | Domains | | | | | | | | | | | |
| | | KP | | | TAP | | | WSP | | | CVRP | | |
| | | Satisfaction | Desire to Complain | N | Satisfaction | Desire to Complain | N | Satisfaction | Desire to Complain | N | Satisfaction | Desire to Complain | N |
| Before Exp. | - | 1.52 (0.89) | 4.32 (1.07) | 100 | 1.79 (1.25) | 3.82 (1.35) | 84 | 1.14 (0.46) | 4.42 (0.97) | 85 | 2.14 (0.97) | 3.89 (0.95) | 76 |
| After Exp. | Baseline | 1.58 (0.84) | 4.11 (1.11) | 36 | 1.63 (0.72) | 4.00 (1.23) | 30 | 1.40 (0.75) | 4.15 (1.19) | 32 | 2.04 (0.93) | 3.64 (1.22) | 25 |
| | Abstract | 2.20 (0.98) | 3.79 (0.90) | 29 | 2.70 (1.06) | 2.67 (1.30) | 27 | 2.18 (1.00) | 3.77 (1.08) | 27 | 2.63 (1.09) | 3.29 (1.16) | 24 |
| | Full | 2.77 (0.94) | 2.97 (1.04) | 35 | 2.59 (0.97) | 2.89 (1.08) | 27 | 2.54 (0.99) | 3.42 (1.14) | 26 | 2.92 (0.87) | 2.85 (1.06) | 27 |
| | Total | 2.18 (0.92) | 3.62 (1.01) | 100 | 2.30 (0.92) | 3.18 (1.20) | 84 | 2.04 (0.91) | 3.78 (1.14) | 85 | 2.53 (0.97) | 3.26 (1.14) | 76 |

Table 2: User's satisfaction (mean (std)) with each explanation in the four domains. $N$ is the number of participants per session.

| Exp. | Domains | | | | | | | |
| | KP | | TAP | | WSP | | CVRP | |
| | $\mu(std)$ | N | $\mu(std)$ | N | $\mu(std)$ | N | $\mu(std)$ | N |
| Baseline | 1.64 (0.96) | 36 | 1.97 (1.22) | 30 | 1.65 (0.97) | 32 | 2.16 (1.07) | 25 |
| Abstract | 2.95 (1.15) | 29 | 2.92 (1.35) | 27 | 2.41 (1.22) | 27 | 2.87 (1.29) | 24 |
| Full | 3.17 (1.22) | 35 | 3.00 (0.91) | 27 | 3.23 (1.17) | 26 | 3.63 (1.08) | 27 |
| Total | 2.58 (1.11) | 100 | 2.62 (1.16) | 84 | 2.43 (1.12) | 85 | 2.89 (1.15) | 76 |

nation. These results are aligned with previous works on social sciences and marketing (Ramon et al. 2021).

## User Study 2: Contrastive vs Counterfactual Explanations

In this user study, we compare the contrastive explanation generated by CMAoE with the counterfactual explanation generated by (Korikov and Beck 2021) (will be referred to as KORIKOV21). To the best of our knowledge, this is the closest state-of-the-art approach that can address OPs.

### Setup

For this comparison, **full** explanations by Q-CMAoE is compared against the KORIKOV21 explanations for KP and WAP domains. As discussed in Related Work, the KO-RIKOV21 can generate counterfactual explanations highlighting a set of hypothetical facts that would have satisfied user's desired chracteristics. Such hypothetical facts are a set of changes that the user could have done in order to get their desired chracteristics. The KORIKOV21 is restricted to specific optimization formulation where the preferences of each user can only be reflected in the objective function and not in the constraints. TAP does not fit this formulation, thus KO-RIKOV21 cannot generate explanations for this domain. In the case of CVRP, the counterfactual explanations would include suggestions to change the distance between the points on the map. From a practical point of view such suggestions are not actionable and realistic and this has been the reason that we did not include this domain in the user study.

For KP and WAP domains, we have generated four sceanrios were at each scenario the initial part of User Study 1 was repeated. At each scerario, each particpant was asked to rate on a 5-point Likert scale, their satisfaction and desire to complain regarding the unfavourable solution for each domain. Regardless of the level of satisfaction or their desire to complain, the partcipants were presented with one of the explanations generated by either CMAoE or KORIKOV21. Afterward, the same set of questions was repeated. At the end of each scneario, participants were asked to rate on a 5-point Likert scale the statements presented in Appendix D regarding explanations. These are metrics adapted from standard ones (Hoffman et al. 2018). Below are the hypotheses that we wanted to test in this user study.

**Hp1**: CMAoE's explanations improve humans' satisfaction more than KORIKOV21's explanations.

**Hp2**: CMAoE's explanations reduce humans' desire to complain more than KORIKOV21's explanations.

**Hp3**: CMAoE's explanations lead to higher scores on explanation goodness metrics than KORIKOV21's explanations.

For this user study, we recruited 40 computer science students or graduates, 10 females and 30 males, average age 27.9 (std=7.27).

### Human Satisfaction & Desire to Complain

THe first part of the user study evaluates *Hp1* and *Hp2*. Table 3 presents the number of users ($N$) and the mean and standard deviation of user satisfaction with the solution and their desire to complain. From a post-hoc analysis with correction for multiple measurements, the decrease in the desire to complain in the KP domain was significantly greater following explanations generated by CMAoE compared to the ones generated by the KORIKOV21 ($p < 0.05$). However, this decrease was not significant for the WAP domain. A detail ANOVA analysis of these results is presented in Appendix E.

These results confirm *Hp1*, however *Hp2* is partially satisfied as humans prefer or equally prefer the explanations generated by CMAoE in comparison to the ones generated by KORIKOV21.

### Good Metric Analysis of Explanations

Using a Univariate Analysis of Variance, we analysed the results of the user study comparing the users' satisfaction with the two types of explanations, using a set of statements adopted from (Hoffman et al. 2018).

Table 3: Satisfaction and desire to complain (mean (std)) about the original solution before and after participants are presented with CMAoE's and Korikov21's explanations in the two domains. $N$ represents the number of participants per session.

| Status | Exp. | Domains | | | | | |
|---|---|---|---|---|---|---|---|
| | | KP | | | WSP | | |
| | | Satisfaction | Desire to Complain | N | Satisfaction | Desire to Complain | N |
| Before Exp. | - | 1.37 (0.83) | 4.42 (0.83) | 40 | 3.23 (0.92) | 3.27 (0.88) | 42 |
| After Exp. | CMAoE | 2.68 (1.01) | 3.05 (1.22) | 19 | 3.09 (1.06) | 2.59 (1.05) | 22 |
| | Korikov21 | 2.19 (1.08) | 3.81 (1.03) | 21 | 3 (1.26) | 2.60 (1.05) | 20 |
| | Total | 2.43 (1.06) | 3.45 (1.18) | 40 | 3.05 (1.15) | 2.60 (1.04) | 42 |

The set of statements and the results are presented in detail in Appendix F. The statistical test was performed with a significance level of 0.05. Based on this, for the Knapsack domain, we found a significant effect for the type of explanation in each of the statements we asked ($p \ll 0.05$). On the other hand, in the WAP domain, despite CMAoE explanations got higher value scores than the Korikov21, their difference were not statistically significant.

These results confirm *Hp3*, where we assumed the participants prefer CMAoE's explanations to Korikov21's ones.

## Related Work

Most works on generating explanations for optimization models focus on explaining infeasibility, often through identifying a minimal (Parker and Ryan 1996; Chinneck 2007) or user preferred (Junker 2004) set of constraints that should be relaxed to get a solution. More recent works also cover optimality in their explanations using different approaches.

(Korikov, Shleyfman, and Beck 2021; Korikov and Beck 2021) propose to use counterfactual explanations for OPs. Given a set of facts and a solution that does not satisfy some desired features, they solve an inverse optimization problem to generate explanations in the form of a hypothetical set of facts that would have satisfied the users' characteristics. Their setting is similar to ours, allowing an individual to inquire about any change to a decision that can be represented with a constraint set on the original formulation. However, while their explanations involve hypothetical features or facts that would yield the user desired output, our explanations highlight the losses incurred in satisfying users' characteristics. Another difference is that their explanation focuses only on the individual asking the question, while our explanation focuses on the rest of the agents involved in the optimization problem. On the experimental side, they limit their evaluation to simulations in two well-known domains, but do not test the validity and usefulness of their explanations with user studies as we do here.

Other literature focuses on contrastive explanations. Cyras *et al.*(2019) explain schedules using argumentation frameworks. In order to provide the explanations, they manually generate the attack graphs, i.e., the relationship between the preferences and the assignments, while we do not need any external input other than the original model and the user's request. A key difference between these works and ours is that they are restricted to makespan scheduling problems with a limited number of preferences, while our approach can provide explanations for any CMAOP. On the evaluation side, they do not report any experiments. Pozanco *et al.* (Pozanco et al. 2022) proposed the EXPRES framework, which also focuses on explaining why the original solution is better than one where the user inquiry is satisfied. However, they: (i) are restricted to linear programs where a totally ordered set of preferences is defined; (ii) need external inputs as in (Cyras et al. 2019); and (iii) only conducted experiments in a workforce scheduling domain, where they measured whether humans preferred explanations generated by other humans or those generated by EXPRES. In this paper, we have conducted experiments in many different CMAOPs, showing how humans' satisfaction with the original solution increased after receiving the explanations generated by CMAoE.

More recently, (Vasileiou, Xu, and Yeoh 2023) proposed QUERIES, a logic-based explanation generation framework that produces both reason-seeking (contrastive) and modification-seeking (counterfactual) explanations that optimize for privacy. In their evaluation, the authors prove that individuals prefer explanations containing only public information, i.e., constraints and preferences known by all agents, over explanations including private information. However, they do not compare, as we do in this paper, whether individuals prefer contrastive or counterfactual explanations.

## Conclusion and Future Work

We have introduced CMAoE, a domain-independent approach to generating contrastive explanations of multi-agent optimization solutions. We generate explanations by building a hypothetical optimization problem that (i) forces the user's requested property to be satisfied; and (ii) minimizes the number of changes between the original and the hypothetical solution. Experimental results through a computational evaluation show how CMAoE can scale in generating contrastive explanations for CMAOPs. Finally, an extensive user study in different CMAOPs shows that explanations generated by CMAoE (i) increase humans' satisfaction with the original solution and decrease their desire to complain; and (ii) humans prefer the contrastive explanations generated by CMAoE over the counterfactual explanations generated by state of the art approaches.

Currently, we are only providing one explanation. However, HCMAOPs often have a few optimal solutions. In future work, we would like to characterize each of these solutions to present a set of diverse explanations from which users could choose. Also, we would like to extend CMAoE to consider agents' privacy or fairness.

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
