# OpenReview forum: "Contrastive Explanations of Centralized Multi-agent Optimization Solutions"
_icaps-conference.org/ICAPS/2024/Conference — ICAPS 2024_

### Official Review · Reviewer_yqJv · 2024-01-22

**Significance And Importance:** 2
**Soundness:** 3
**Novelty:** 3
**Clarity:** 4
**Overall Evaluation:** 2
**Confidence:** 5

**Weaknesses:**

1: Minor weaknesses that are easily fixable.

**Contributions Of The Paper:**

The paper revisits the concept of counterfactual explanations in the context of centralized multi-agent optimization problems. In particular, the paper builds and extends the work done by (Krarup et al. 2021) into this new problem space. The evaluation includes both a computational evaluation and user studies. Computational evaluation establishes the scalability of the proposed method, while the user study focuses on subjective aspects.

**Ethical Considerations:**

(3) Fair: The paper addresses some applicable ethical considerations but fails to address some important ones

**Nomination For Best Paper:**

No

**Questions For Authors:**

Please respond to my assertions about

1. Whether an agent receives the explanation needs to understand the compilation
2. The limits on the kinds of similarity supported by the method
3. The limits on the kind of properties and, by extension, the questions that can be asked by the user.

I would also appreciate a response on the question regarding the need for multiple agents, but it is not a requirement

**Reproducibility:**

3: Authors describe the implementation and domains in sufficient detail.

**Strengths Of The Paper:**

The idea to bring ideas originally introduced in the context of planning literature into other problem settings, where the methods could be applied in much larger problem instances. It is always nice to see XAI papers with extensive user studies.

**Weaknesses Of The Paper:**

Below are some of the questions and comments I had about the paper (I will also repeat them in the question section)

Evaluation - While I applaud the authors' decisions to run user studies to evaluate the method. Seeing that the evaluation was just focused on a subjective metric is a bit of a missed opportunity. It is usually better to evaluate both subjective aspects regarding the method (acceptance, preference, etc..), along with the objective measures like how well the explanation may help the user better understand the task or help improve their ability to complete the task or make better decisions. I would highly encourage the authors to look at [1] for more guidelines on evaluating explanation generation methods.

Explanations themselves - It seems the explanations provided here actually compare the values taken by decision variables. This somehow implies the agents have access to and can understand the MIP encoding being used here. This doesn’t seem very realistic. While it works in these simpler problem settings, I would imagine more complex problems may require more involved compilations with multiple non-obvious decision variables. How well would an agent who is not an expert in MIP compilations be able to make sense of such explanations?

Distance measures and properties - Finally, it seems the paper currently focuses on a rather simple distance measure, namely the difference in values. I would imagine there are cases where similarity or dissimilarity of solutions could involve more complex notions (for example, there may be some qualitative notions of similarity, like how certain agents are favored over others, which might be hard to capture through just the difference in the value of individual variables). Also, I am assuming the fact that properties need to be encoded into MIP also constrains the kind of questions the user can ask.

On a smaller note/question, how important is the fact that this is a multi-agent optimization problem? Would any algorithmic mechanism change if it were a traditional optimization problem?

[1] Hoffman, Robert R., et al. "Metrics for explainable AI: Challenges and prospects." arXiv preprint arXiv:1812.04608 (2018).

---

> ### Author Rebuttal · Authors · 2024-01-26
>
> We completely agree with the reviewer, and indeed, the Hp3 in User Study 2 focuses on the goodness metrics adopted from Hoffman et al.. The set of statements and the results are presented in detail in Appendix F. As suggested by the reviewer, we believe statements 4-7 aim to evaluate the objective measures. We will move more details of these results to the main body of the paper.
> Regarding the agents' understanding of the MIP, the only implicit assumption we are making is that the agents understand the solution without the need to understand what the variables are that were used to represent a solution nor the auxiliary decision variables that are used to model the problem. In case some agents have a deeper understanding of the MIP, they would also be able to ask questions about the value of these auxiliary variables. Even if our current distance measure might be regarded as simplistic, it is definitely useful in practice, as reducing the number of changes between two solutions can arguably help agents understand the differences between the initial solution and the one they are asking for. Exploring alternative distance measures is definitely an interesting topic that we would like to address in future work.
>
> Regarding the limitations for the properties, as long as the user is able to form their question in the form of ”why does solution S not satisfy property P?”, CMAOE is able to generate a solution for the task. Naturally, we assume that the users are only able to ask questions regarding properties that are included in the optimisation problem.
> Regarding the multi-agent setting, we have focused on multi-agent settings where the decision-making is centralized (the central entity is solving the optimization task, which involves multiple agents). The central entity needs to consider each agent’s preferences and properties in the decision-making process. Our explanations aim to help explain such decisions to both the central entity and the agents involved. We believe this is the kind of scenario where explanations can be very useful. There are many real-world scenarios that lie under this framework, such as nurse shift scheduling, task, or resource allocation. In the nurse scheduling problem, these explanations can be used for the person assigning the shifts to understand why a specific schedule is assigned and also for the nurses who have received a schedule and would like to understand why the AI did not assign them a different schedule.

---

### Official Review · Reviewer_kBTM · 2024-01-23

**Significance And Importance:** 2
**Soundness:** 3
**Novelty:** 2
**Clarity:** 3
**Confidence:** 3

**Weaknesses:**

-1: Major weaknesses requiring significant work to be addressed for the paper to be accepted.

**Contributions Of The Paper:**

This work explores contrastive explanations for multi-agent (not planning) problems by parsing them to mixed integer programs. It incorporates the conditions to the integer program to find a new solution and compare it to provide the explanation. They evaluate the explanation generation for scalability and solution quality. They also evaluated the generated explanations through user study for satisfaction.

**Ethical Considerations:**

(4) Good: The paper adequately addresses most, but not all, of the applicable ethical considerations

**Nomination For Best Paper:**

No

**Overall Evaluation:**

-1: (weak reject)

**Questions For Authors:**

(1) Are there any limitations what kind of explanation properties the user can query about. Can all the property be easily converted to representable constraint set?
(2) The domains look a lot more combinatorial problems, instead of planning problems. Was that intentional? If so, compared to [1] (in Weakness section) is your solution approach more generic or more specialized. What is the difference in application of your approach to the paper that inspired the approach.

**Reproducibility:**

3: Authors describe the implementation and domains in sufficient detail.

**Strengths Of The Paper:**

The paper is well-written and has a thorough evaluation with multiple user studies involving more than 200 people.

**Weaknesses Of The Paper:**

I have two questions/issues with the paper --

(1) The paper is motivated towards multi-agent scenarios, which should mean that there is parallelism and concurrency between various agents in the scenario. The explanations should be oriented towards the multi-agent scenarios (or multi-agent oriented properties). However, in the current case the evaluation of scalability, quality and the user study is about whether some property changes and the explanation is provided. The properties are about specific parts of the solution and has no relevance to multi-agents. The problem solution is centralized, the explanation and to some extent the domains have no relation to the multi-agent scenarios either. It also follows the template of contrastive explanation which is one solution compared with another solution generated after constraining based on the property queried by the user. Thus, I am not sure what is the use of multi-agent optimization problems.
(2) Currently, the work needs more detail for the user study, most importantly the procedure followed to conduct the user study. It's unclear what was shown to the user, what properties they could question, how was the interface, etc. These details are necessary in the main paper in order to understand what was the real user study. Does it really gives the results for user satisfaction or not.

Other suggestions --
(2) The paper is missing comparison with other explanation related user studies that have been conducted. You do cite the Krarup's paper [1], but have no comparison as to why your contrast based explanation are different. In the current state I believe your work is just another way of implementation of the template for contrastive explanation. You also need to compare whether there technique can work for you, and if so then how does your solution strategy differ, and what novelty it brings to the table. You also miss the user study in [2]. I would love to see your work in compared and contrasted to user study specific papers as well. They should also be discussed in related work.

References --
[1] Krarup, B., Krivic, S., Magazzeni, D., Long, D., Cashmore, M. and Smith, D.E., 2021. Contrastive explanations of plans through model restrictions. Journal of Artificial Intelligence Research, 72, pp.533-612.
[2] Chakraborti, T., Sreedharan, S., Grover, S. and Kambhampati, S., 2019, March. Plan explanations as model reconciliation--an empirical study. In 2019 14th ACM/IEEE International Conference on Human-Robot Interaction (HRI) (pp. 258-266). Ieee.

---

> ### Author Rebuttal · Authors · 2024-01-26
>
> Regarding the questions about multi-agent setting and type of properties that can be questioned by the user please refer to reviewers 3 responses.
> Our focus is on combinatorial problems rather than on planning tasks. Note that both fields are very related, and works on combinatorial problems, optimization and scheduling are often accepted at ICAPS. Regarding the user study; we have recruited a number of students (demographic data is reported in the paper) that were all volunteered. An expert presented the users with a toy OP example to explain the process of the user study. Then, the users have filled a form, specifying their demographic data. The users used an online website which presented them the scenarios and their solution. Next, they were asked questions regarding each scenario to ensure they have understood the scenarios correctly. The participants who failed at this stage were removed from the user study. At the end each user has been asked with a series of questions shown in Appendix to evaluate their satisfaction with the solution/explanations and the comparison against the KORIKOV21.
> Regarding the related works, as it is mentioned in the introduction, our work is inspired by Krarup et al., where the focus is on local contrastive explanations by generating a new plan that satisfies the given property, and answers the question based on comparing the original and the hypothetical plan. There exist two main differences. First, [1] focus on planning tasks, while we focus on optimization problems. Although planning techniques can be used to solve optimisation problems and vice versa, the modelling and performance gap can be huge in many cases, leaving one of them as the most suitable approach. The same applies to [2], which focuses on planning tasks rather than optimisation problems. Second and more importantly, they do not focus on minimizing the number of changes (length of the explanation) between the original and the hypothetical plans, while this is a core feature in our approach.

---

### Official Review · Reviewer_B2M7 · 2024-01-23

**Significance And Importance:** 2
**Soundness:** 3
**Novelty:** 2
**Clarity:** 4
**Overall Evaluation:** 1
**Confidence:** 3

**Weaknesses:**

2: No major or minor weaknesses.

**Contributions Of The Paper:**

The article proposes a method, CMAoE,  to give contrastive explanations about solutions to optimization problems. The explanation aims to answer the counterfactual question: "Why does solution S not satisfy property P?".

CMAoE is based on a hypothetical optimization problem that combines the problem and the original solution, and counterfactual property P. The explanation contrasts the original solution with the solution of the hypothetical problem.

CMAoE is formalized in Integer Programming (IP) problems and experiments are carried out on 4 traditional PI problems.

In addition to evaluating the quality of explanations with objective metrics: explanation size and loss of quality, experiments are also carried out with users. Finally, CMAoE is compared with a method based on counterfactual explanation from recent literature.

**Ethical Considerations:**

(1) Not Applicable: The paper does not have any ethical considerations to address

**Nomination For Best Paper:**

No

**Questions For Authors:**

1 -  In the following paragraph it seems that is missing the formal constraint.

"First, we enforce the agent’s property by adding the following constraint, which ensures that the bed is included in the depot."

2 - I missed in the paper the concrete explanation given to the user. Only by checking in the Supplemtary Material I could find such a information.
For example, abstract information will always be: 'Total utility would decrease'?

Also, it would be good to have the concrete explanation given my the Korikov21 method.

3 - In the objective function after line 260, it seems that x_{a,i} should be x'_{a,i}.

4 - In some parts of the paper, WSP is misspelled WAP.

**Reproducibility:**

3: Authors describe the implementation and domains in sufficient detail.

**Strengths Of The Paper:**

- Experiments with user

- the formalization of the problem

**Weaknesses Of The Paper:**

- experiments are done only with synthetic problems

---

> ### Author Rebuttal · Authors · 2024-01-26
>
> We will address all the points the reviewer has raised. Regarding the second question raised by the reviewer, due to the space constraints we had left the explanations generated by CMAOE in Supplementary Material. However, we will ensure to include an example of our explanation for the running example to the body of the paper and refer the reader for more details to the Appendix. Regarding our explanations, we assume that: (a) the initial solution provided to the agent is optimal (based on the objective function that is considered in optimization); and (b) we are not considering the scenarios where the problem has several optimal solutions. Thus, we can expect that all the explanations generated by CMAOE will be of the form ”Total utility would decrease by X” where X is the value returned by the algorithm. Unfortunately, we were not able to find any experimental evaluation and/or examples of the explanations generated by KORIKOV21. However, in User Study 2, we have generated explanations based on the approach suggested by KORIKOV21. We will add an example of the scenario we ran in this user study in Appendix.

---

### Meta-Review · Area_Chair_89vp · 2024-02-07

**Recommendation:** Accept (Poster)
**Confidence:** 4

**Metareview:**

This paper introduces a technique for generating contrastive explanations of centralized multiagent optimization problems. It draws inspiration form earlier work in generating similar explanations for planning problems, and given an optimal solution, generates explanations of the form: “Why does solution S not satisfy property P? The approach is formulated in terms of an Integer Linear Program (ILP) solver and is evaluated over 4 traditional ILP problems. A computational study to analyze objective performance and multiple user studies to evaluate subjective effects are performed and reported. Major results show that humans’ satisfaction with the original solution increases after being presented with explanations and that explanations generated by the proposed approach are preferred by humans over other state of the art approaches.

Strengths: The paper adapts the idea of generating counterfactual explanations for planning problems to a different class of larger, multiagent optimization problems and establishes its broader applicability. The utility of the proposed technique is validated through extensive user studies. User studies are also used to validate how explanation length affects how the user perceives the explanation. Finally, the paper is well-written and mostly easy to follow.

Weaknesses: First, the paper does not clearly define what is meant by the term “centralized multiagent optimization problems”. The 4 problems considered in the experimental analysis are classical combinatorial optimization problems, but in what sense are they multiagent problems? From comments in the discussion, you seem to be referring to scheduling and allocation problems that involve assignment of resources (agents) to tasks over time. But this problem class is never explicitly defined, and this adds unnecessary confusion to the paper’s focus. More importantly, the paper does not do a good job of characterizing the unique challenges associated with applying contrastive explanations to this different class of problems. Why are these problems different and/or more difficult to explain than planning problems? The paper needs to better motivate the significance of the work and its contribution. Second, some of material needed to fully understand the paper is contained only in the appendices that were provide as supplementary materials and this detracts considerably from the paper’s comprehensibility in different spots. The author(s) responses to reviewer questions indicate their intention to move information essential to the paper’s claims and arguments into the paper itself, as well as provide a running example that shows what types of explanations are generated and considered in the user studies. Such revisions are seen as necessary for the paper to be accepted. Finally, the paper is missing any comparison with other contemporary explanation techniques that have carried out user studies. Minimally these should be mentioned when discussing related work.

**Ethical Considerations:**

(4) Good: The paper adequately addresses most, but not all, of the applicable ethical considerations